# Dysbiosis of gut microbiota in C57BL/6-Lep[em1hwl]/Korl mice during microplastics-caused hepatic metabolism disruption

Yu Sang Choi[1][☯], Yu Jeong Roh[1][☯], Ji Eun Kim[1], Hee Jin Song[1], Ayun Seol[1], Su Jeong Lim[1], Su Ha Wang[1], Ji Eun Sung[1], Ye Eun Ryu[1], Ye Ryeong Kim[1], Hyun Su Park[1], Hong Joo Son[2], Yu Jin Kim[2], Eun Sang Jung[3], Sunho Park[4], Dae Youn Hwang[1]*

1 Department of Biomaterials Science (BK21 FOUR Program)/Life and Industry Convergence Research Institute/Laboratory Animal Resources Center, College of Natural Resources & Life Science, Pusan National University, Miryang, Korea, 2 Department of Life Science & Environmental Biochemistry/Life and Industry Convergence Research Institute, College of Natural Resources & Life Science, Pusan National University, Miryang, Korea, 3 Department of Bioenvironmental Energy/Life and Industry Convergence Research Institute, College of Natural Resources & Life Science, Pusan National University, Miryang, Korea, 4 Department of Bio-Industrial Machinery Engineering/Life and Industry Convergence Research Institute, College of Natural Resources & Life Science, Pusan National University, Miryang, Korea

☯ These authors contributed equally to this work.
* dyhwang@pusan.ac.kr

## Abstract

The oral administration of microplastics (MP) for 9 weeks induced disruption of hepatic lipid, glucose, and amino acid metabolism in C57BL/6-Lep[em1hwl]/Korl (Lep KO) mice with obesity. Therefore, we investigated whether MP-caused hepatic metabolism abnormalities can affect the structural variation of the fecal microbiota during obesity. The overall microbiota profile was analyzed in the feces of Lep KO mice treated with MP for 9 weeks. The lipid accumulation and steatosis area were significantly decreased in MP-treated Lep KO mice. Total microbiota with MP-caused difference identified from feces of Lep KO and wild type (WT) mice were classified into 10 phyla and 106 genera. Among them, two microbial phyla were significantly changed in Lep KO mice after treatment of MP, while significant alterations on 12 genera were detected in Lep KO mice treated with MP. Also, the Chao1 index for richness were remarkably decreased in both MP-treated Lep KO and WT mice, but Shannon index for evenness were increased in only MP-treated Lep KO mice. Therefore, the results of the present study suggest that MP-caused hepatic metabolism disruption may be closely linked to the dysbiosis of the fecal microbiota in Lep KO mice.

## Introduction

The microbiota consists of microorganisms and their genes residing in the intestinal tract, performing various roles in the host's body [1]. It promotes the digestion of dietary carbohydrates, lipids, and proteins and contributes to the development

**Data availability statement:** All relevant data are within the manuscript and its Supporting information files.

**Funding:** This work was supported by the BK21 FOUR Program through the National Research Foundation of Korea (NRF), funded by the Ministry of Education (F25YY8109033 to D.Y.H.), Republic of Korea. Also, this work was supported by 2025 Specialization Project of Pusan National University (SG-2025-006 to D.Y.H).

**Competing interests:** No authors have competing interests.

of gastrointestinal mucus and the maturation or activation of cells in the intestinal immune system [2–5]. Recent studies have suggested a correlationship between the hepatic metabolism and changes in the microbiota. The gut–liver axis is a bidirectional link, with many metabolites transported from the gastrointestinal tract to the liver via the hepatic portal vein [6]. Therefore, the hepatic metabolism can be affected by gut dysbiosis and cause insulin resistance, hepatic steatosis, and cirrhosis [7]. In a previous study, patients with metabolic-associated fatty liver disease showed an altered gut microbiota compared to healthy individuals [8]. The administration of the probiotic VSL#3 into ApoE$^{-/-}$ mice, a genetic model of dyslipidemia, intestinal inflammation, and steatohepatitis, showed improvement of insulin resistance and steatohepatitis [9]. These results provide some scientific evidances for the bidirectional alterations between the gut microbiota and hepatic metabolism.

Polystyrene microplastics (PS-MP) are considered a major factor influencing the gut microbiota composition despite differences in treatment conditions, including their size, treatment period and routes, and animal strain. The treatment of ICR mice with PS-MP for five to six weeks induced alterations in the gut microbiota at the phylum and genus levels during hepatic metabolism disorder [10,11]. In addition, similar changes in the gut microbiota and disruption of the liver metabolism, including inflammation and fatty acid synthesis, were detected in the PS-MP (20–5 µm)-treated C57BL/6 mice or BALB/c mice for 28–60 days [12–16]. The correlation between neuronal damage and dysbiosis of the gut microbiota was detected in C57BL/6 mice orally administered < 5 mm PS-MP for 30 days [17]. Furthermore, the disruptions in the richness and diversity of gut microbiota at the phylum and genus levels were observed in C57BL/6 mice when hepatotoxic or colitis-inducing compounds and PS-MP were administered simultaneously [18–20]. Despite this, no studies have examined whether MP-caused abnormalities in the hepatic metabolism can affect the microbial composition of the gut in leptin-deficient models with obesity phenotypes after long-term MP treatment.

This study characterized the structural variations of fecal microbiota in C57BL/6-Lep$^{em1hwl}$/Korl (Lep KO) mice, which experienced metabolic disorders caused by the administration of PS-MP for nine weeks.

## Materials and methods

### Management of animals

The Pusan National University-Institutional Animal Care and Use Committee (PNU-IACUC) approved the protocols of the animal experiments based on ethical procedures for scientific care (Approval Number PNU-2022-0191 and PNU-2023-0350). All efforts were made to minimize the pain during administration of MP and other experimental process. Mice were euthanized by trained researchers using $CO_2$ gas with a minimum purity of 99.0% according to the American Veterinary Medical Association (AVMA) Guidelines for the Euthanasia of Animals. Their final death was confirmed by various physiological signs including cardiac and respiratory arrest, or dilated pupils and a fixed body. All experiments related to mice were conducted at the PNU-Laboratory Animal Resources Center, accredited by the Ministry of Food and Drug Safety (MFDS) (Accredited Unit Number-000231) and the Association for Assessment

and Accreditation of Laboratory Animal Care (AAALAC) International (Accredited Unit Number; 001525). Filtered tap water and a standard diet (Samtako Bio Korea Co., Osan, Korea) comprising crude protein (22.5%), fat (5.0%), fiber (4.5%), ash (6.5%), nitrogen-free extracts (50.5%), and moisture (11.0%) were offered *ad libitum*. The mice were maintained in a specific pathogen-free (SPF) state under a light-dark cycle (08:00 h to 20:00 h) at 23 ± 2°C and a relative humidity of 50 ± 10%.

## Design of animal experiment and collection of tissue samples

The animal experiments were carried out using the method reported elsewhere [21]. Laboratory Animals Banks in the Korea Food and Drug Administration (KFDA) kindly provided the WT and Lep KO mice with C57BL/6 background strain. The WT (n=12) and Lep KO mice (n=12), aged seven weeks, were subdivided into two subgroups Vehicle-treated WT mice (n=6), MP-treated WT mice (n=6), Vehicle-treated Lep KO mice (n=6), and MP-treated Lep KO mice (n=6). To ensure the reliability of the data from the animal study, the total number of animals was determined as 24 using G-POWER 3.1.9.7 (Heinrich-Heine-Universität Düsseldorf, Germany) with the α error probability of 0.05, effect size of 0.9 and a power of 0.95. The Vehicle-treated groups were orally administered 0.5 mL of 1× phosphate-buffered saline (PBS) solution once a day, three days per week, for nine weeks. The MP-treated groups were also orally administered the same volume of PS-MP (0.5 µm) suspension dispersed at 100 µg/mL for the same duration. The PS-MP dosage was determined based on a previous study [22]. At twenty-four hours after the final administration, fresh feces and liver tissues from subset groups of WT and Lep KO mice were collected using metabolic cage (Fig 1A). All fecal and liver samples were stored at −80°C until further analysis.

## Detection of MP accumulation

The liver tissues collected from WT and Lep KO mice of subset groups were soaked in optimal cutting temperature (OCT) compound to facilitate cryoprotection. The tissue blocks were frozen at −80°C for 24 h. Cryosection was performed using a cryostat to obtain approximately 20 µm thick sections. The sections were adhered to glass slides. The fluorescence of MP was detected under a fluorescence microscope at 100× and 400× magnification.

## Histopathological analyses

Histopathology analysis of the liver tissue was performed as described elsewhere [21]. Briefly, the liver tissues (500 mg) were fixed in 10% formalin solution for 48 h, followed by trimming and embedding in paraffin wax. The embedded tissues were sliced into 4 µm thick sections and stained with hematoxylin and eosin solution (H&E; Sigma–Aldrich Co.). After mounting the tissue section, the morphological features of liver tissue were observed by optical microscopy and analyzed using the Leica Application Suite (Leica Microsystems, Heerbrugg, Switzerland). H&E-stained slides were examined for three to five samples per group under 100× and 400× magnification.

## 16S rRNA sequencing

The overall analyses of fecal microbiota were conducted as outlined in a previous study [23]. Briefly, the feces collected from each mouse were pooled into two groups for this analysis. The total DNA was collected from the fresh feces (1 g) of each pool using a DNeasy PowerSoil Kit (Qiagen, Hilden, Germany). The sequencing libraries were prepared according to the 16S Library Preparation Protocol. The 16S rRNA gene products were amplified using a two-step PCR process with Nextera XT Index primers and purified based on the qPCR Quantification Protocol Guide (KAPA Library Quantification Kits for Illumina sequencing platforms).

## Analysis of sequencing data

The sequences of PCR products were analyzed using the Macrogen unit on the MiSeq™ platform (Illumina, San Diego, CA, USA). The data were curated using the FASTQ program [24] and assigned to Operational Taxonomic Units (OTUs)

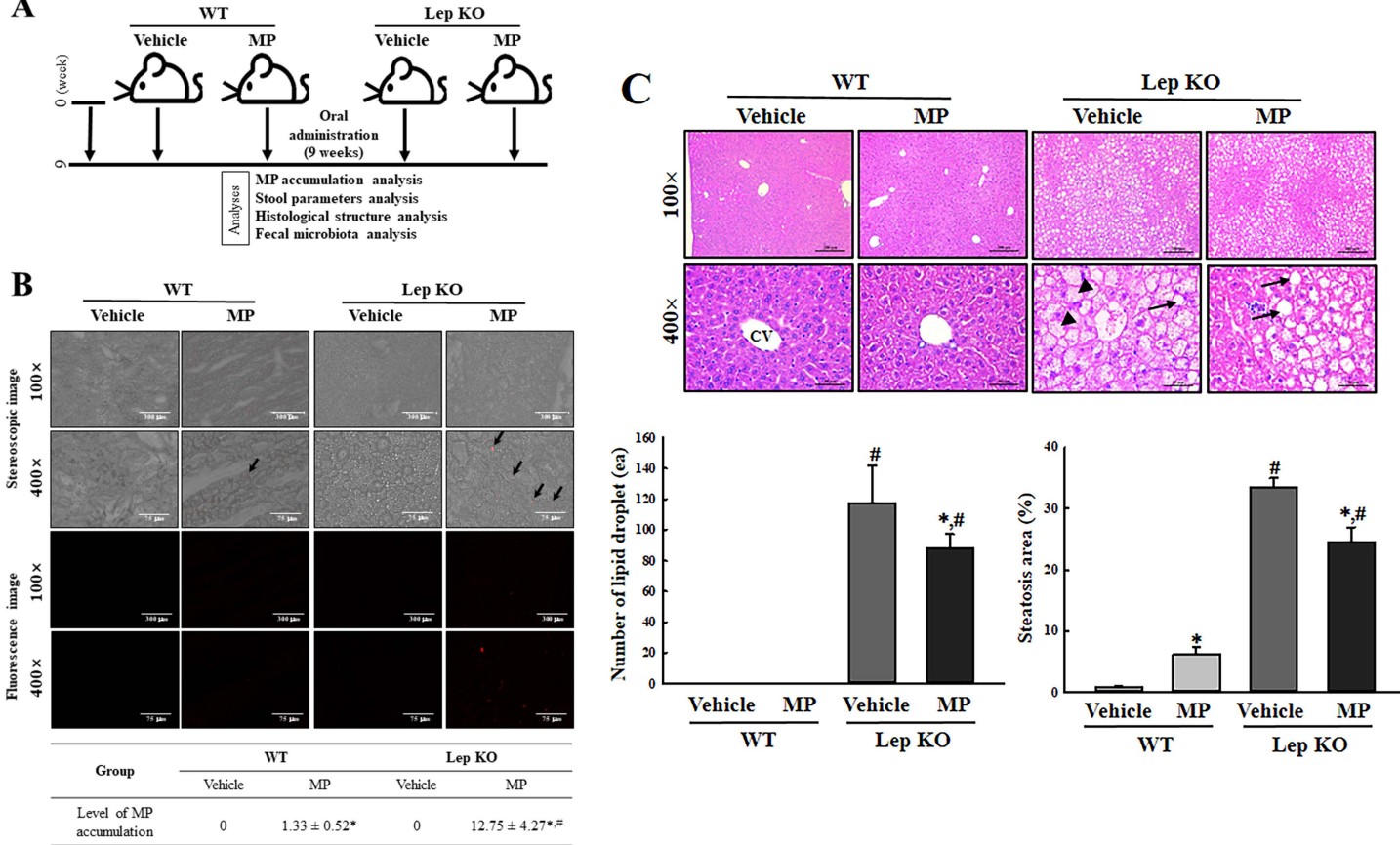

**Fig 1. Disruption of hepatic metabolism in Lep KO mice. (a)** Experimental scheme. PS-MP was orally administrated into WT and Lep KO mice for 9 weeks. **(b)** Accumulation of MP in liver tissue. The red fluorescence of MP was detected under a fluorescence microscope at 100× and 400× magnification. Arrows indicate MP. The frozen sections of liver tissues were prepared in triplicates of each mice, and each section was analyzed twice. **(c)** Histopathological structure of the liver. Pathological alterations were analyzed in H&E-stained liver sections at 100× and 400× magnification. Arrow heads indicate microvesicular steatosis and arrows indicate macrovesicular steatosis. The H&E-stained sections of liver tissues were prepared in triplicates of each mice, and their histopathological factors were analyzed twice per each section. The data are reported as mean±SD values. *, $p < 0.05$ compared to the Vehicle-treated group. #, $p < 0.05$ compared to the WT mice. Abbreviation: WT; wild type, Lep KO; leptin knockout, MP; microplastics, H&E; hematoxylin and eosin, CV; central vein.

using the Cluster Database at High Identity with Tolerance (CD-HIT) [25]. The sequences of each OTU were aligned according to BLAST+ (v2.9.0) [26] in the Reference DB (NCBI 16S Microbial) and analyzed using the MAFFT (v7.475) program. Comparative analysis of various microbial communities was conducted using QIIME (v1.9) [27]. The α-diversity information was obtained using the Shannon index, and the species richness was estimated using the Chao1 index. The unweighted and weighted UniFrac distances were used to assess the β-diversity. Principal coordinate analysis (PCoA) was used to illustrate the dissimilarity between the microbiota [27].

## Statistical analysis

The statistical significance of each experimental group was determined using the one-way analysis of variance (ANOVA) or the unpaired two-sample t-test. After assessing the normality using a Shapiro–Wilk test in SPSS 27.0 (IBM Co., Armonk, NY, USA), Tukey multiple comparisons tests were also conducted to support these two analyses. All values used in this study are reported as the means±SD; $p$ values <0.05 were considered significant.

## Results

### Confirmation of MP and fat accumulation in the liver

First, this study confirmed whether the administration of MP can disrupt the hepatic metabolism to ensure that it is suitable for microbiota analysis. The fluorescence intensity of the MP and histological structure was observed in a frozen and H&E stained section of the liver tissue of Lep KO mice treated with MP for nine weeks under a fluorescence microscope and optical microscope. MP particles were detected in the liver section of the MP-treated Lep KO mice, while nothing was detected in the Vehicle-treated Lep KO mice. Their number was higher in MP-treated Lep KO mice than MP-treated WT mice (Fig 1B). The total number of lipid droplets and the area of steatosis were significantly lower in the liver tissue of the MP-treated Lep KO mice than in the Vehicle-treated Lep KO mice (Fig 1C). These results suggest that Lep KO mice with MP-caused hepatic metabolism disruption are suitable for analyzing the gut microbiota.

### Effects of MP-caused hepatic metabolism disruption on the profile of the fecal microbiota at the phylum level

The overall microbiota profile was examined in fecal samples from MP-treated Lep KO mice to determine if the structural variations of the fecal microbiota at the phylum level are associated with MP-caused hepatic metabolism disruption. Approximately 98.20% of the total microbial sequences isolated from the feces were classified into 10 phyla and 106 genera (Figs 2A and 3A). In addition, a significant alteration in six microbial phyla, including four phyla of WT and two phyla of Lep KO mice, was observed between the Vehicle- and MP-treated mice. Among them, the populations of four phyla (Bacillota, Bacteroidota, Mycoplasmatota, and Verrucomicrobia) were remarkably changed in the MP-treated WT mice compared to the Vehicle-treated WT mice. The most significant changes were Mycoplasmatota (225%), followed by Verrucomicrobia (−100%), Bacillota (19.97%), and Bacteroidota (13.02%). In the case of Lep KO mice, however, significant changes were detected on only two phyla. The populations of *Candidatus Melainabacteria* and Deferribacterota were remarkably increased by 40% and 110%, respectively (Fig 2). These results suggest that a MP-caused hepatic metabolism disruption can contribute to the change in the phylum level of gut microbiota profile in Lep KO mice.

### Effects of MP-caused hepatic metabolism disruption on the profile of the fecal microbiota at the genus level

This study next examined whether the structural variation of the fecal microbiota at the phylum level during MP-caused hepatic metabolism disruption was accompanied by those of the genus level. The overall microbiota profile in the fecal samples from MP-treated Lep KO mice was analyzed at the genus level. Thirty-five microbial genera were different in the Vehicle-treated and MP-treated mice, with 23 genera in the WT and 12 genera in Lep KO mice showing changes (Fig 3A). In particular, the populations of 23 genera, including *Christensenella* (−260%), *Butyricicoccus* (−350%), *Hungatella* (88%), *Eubacterium* (62%), *Acetatifactor* (−183%), *Dorea, Enterocloster* (−88%), *Frisingicoccus* (−100%), *Fusimonas* (700%), *Lachnoclostridium* (131%), *Mediterraneibacter* (187%), *Simiaoa* (1,100%), *Sporofaciens* (222%), *Waltera* (400%), *Acutalibacter* (600%), *Anaerotruncus* (75%), *Angelakisella* (233%), *Oscillibacter* (200%), *Ruminococcus* (−73%), *Sangeribacter* (122%), *Alistipes* (168%), *Anaeroplasma* (225%), and *Akkermansia* (−100%) in the MP-treated WT mice were significantly different from that of the Vehicle-treated WT mice (Fig 3B and 3C). In contrast, significant changes were detected in 12 genera in the Lep KO mice. The most significant changes were *Lactobacillus* (4,233%), followed by *Anaerocolumna* (1,300%), *Jutongia* (333%), *Limosilactobacillus* (227%), *Anaerotaenia* (200%), *Ruminococcus* (200%), *Heminiphilus* (−86%), and *Feifania* (−60%). The populations of *Vampirovibrio, Wansuia, Mageeibacillus*, and *Saccharofermentans* were increased by 40%, 25%, 25%, and 15%, respectively. (Fig 3B and 3C). These results show that the structural variations of the fecal microbiota at the phylum level during PS-MP-caused hepatic metabolism disruption can be linked to their alterations at the genus level.

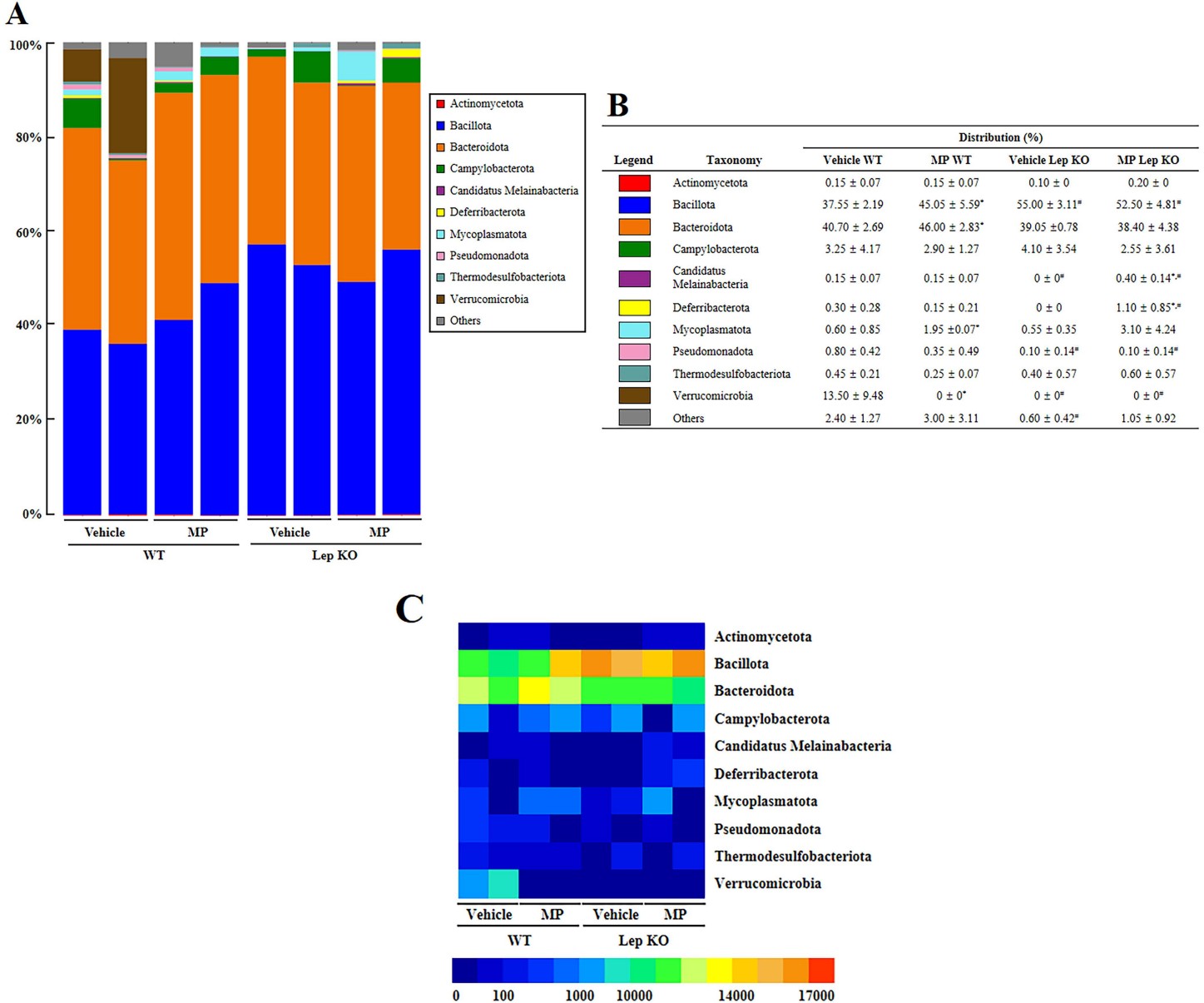

**Fig 2. Composition of the gut microbiota at the phylum level following the administration of MP. (a)** Fecal microbiota distribution at the phylum level in WT and Lep KO mice. **(b)** Relative level of fecal microbiota distribution at the phylum level in WT and Lep KO mice. Each color represents one bacterial phylum. **(c)** Heat map showing a significant difference. Different colors indicate the relative abundance of each phylum. The data are reported as mean ± SD values. *, $p < 0.05$ compared to the Vehicle-treated group. #, $p < 0.05$ compared to the WT mice. Abbreviation: WT; wild type, Lep KO; leptin knockout, MP; microplastics.

### Effects of MP-caused hepatic metabolism disruption on the diversity and dissimilarity of fecal microbiota

Finally, this study investigated whether MP-caused hepatic metabolism disruption can affect the structural diversity and dissimilarity of the fecal microbiota in Lep KO mice. The changes in the richness, evenness, and dissimilarity of the gut microbiota in the feces of MP-treated Lep KO mice were analyzed. The α-diversity was analyzed using the Chao1 index for richness and the Shannon index for evenness. The Chao1 index decreased significantly in the MP-treated WT and

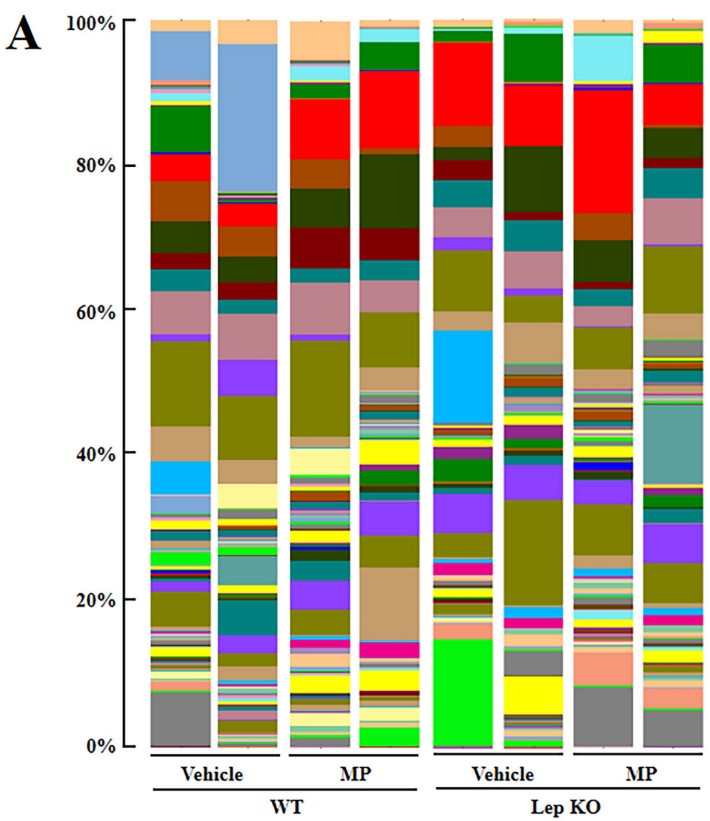

**A**

| | | |
|---|---|---|
| Vehicle | MP | WT |
| Vehicle | MP | Lep KO |

Legend:

- ■ Corynebacterium
- ■ Leptogranulimonas
- ■ Adlercreutzia
- ■ Enteroscipio
- ■ Xiamenia
- ■ Lactobacillus
- ■ Ligilactobacillus
- ■ Limosilactobacillus
- ■ Streptococcus
- ■ Flintibacter
- ■ Intestinimonas
- ■ Natranaerovirga
- ■ Christensenella
- ■ Guopingia
- ■ Luoshenia
- ■ Butyricicoccus
- ■ Clostridium
- ■ Hungatella
- ■ Eubacterium
- ■ Intestinibacillus
- ■ Ihubacter
- ■ Zhenpiania

- ■ Feifania
- ■ Gracilibacter
- ■ Acetatifactor
- ■ Anaerocolumna
- ■ Anaeromicropila
- ■ Anaerosporobacter
- ■ Anaerotaenia
- ■ Anaerotignum
- ■ Blautia
- ■ Coprococcus
- ■ Dorea
- ■ Eisenbergiella
- ■ Enterocloster
- ■ Faecalicatena
- ■ Frisingicoccus
- ■ Fusimonas
- ■ Jingyaoa
- ■ Jutongia
- ■ Kineothrix
- ■ Lachnoclostridium
- ■ Lacrimispora
- ■ Marvinbryantia

- ■ Mediterraneibacter
- ■ Muricomes
- ■ Murimonas
- ■ Qiania
- ■ Roseburia
- ■ Simiaoa
- ■ Sporofaciens
- ■ Tyzzerella
- ■ Variimorphobacter
- ■ Velocimicrobium
- ■ Waltera
- ■ Wansuia
- ■ Acetivibrio
- ■ Acutalibacter
- ■ Anaerobacterium
- ■ Anaerotruncus
- ■ Angelakisella
- ■ Congzhengia
- ■ Fumia
- ■ Harryflintia
- ■ Lawsonibacter
- ■ Mageeibacillus

- ■ Marasmitruncus
- ■ Marseillibacter
- ■ Neglectibacter
- ■ Oscillibacter
- ■ Paludihabitans
- ■ Petroclostridium
- ■ Pseudoflavonifractor
- ■ Ruminiclostridium
- ■ Ruminococcus
- ■ Saccharofermentans
- ■ Vescimonas
- ■ Dehalobacterium
- ■ Vallitalea
- ■ Amedibacillus
- ■ Faecalibaculum
- ■ Longibaculum
- ■ Tannockella
- ■ Turicibacter
- ■ Bacteroides
- ■ Duncaniella
- ■ Heminiphilus
- ■ Muribaculum

- ■ Paramuribaculum
- ■ Sangeribacter
- ■ Culturomica
- ■ Prevotella
- ■ Alistipes
- ■ Tidjanibacter
- ■ Parabacteroides
- ■ Helicobacter
- ■ Vampirovibrio
- ■ Mucispirillum
- ■ Anaeroplasma
- ■ Aestuariispira
- ■ Rhodospirillum
- ■ Parasutterella
- ■ Bilophila
- ■ Desulfovibrio
- ■ Akkermansia
- ■ Others

**B**

| Legend | Taxonomy | Distribution (%) | | | |
|---|---|---|---|---|---|
| | | Vehicle WT | MP WT | Vehicle Lep KO | MP Lep KO |
| | Corynebacterium | 0.05 ± 0.07 | 0.05 ± 0.07 | 0 ± 0 | 0.10 ± 0 |
| | Leptogranulimonas | 0.05 ± 0.07 | 0 ± 0 | 0 ± 0 | 0 ± 0 |
| | Adlercreutzia | 0 ± 0 | 0 ± 0 | 0 ± 0 | 0.05 ± 0.07 |
| | Enteroscipio | 0 ± 0 | 0.05 ± 0.07 | 0 ± 0 | 0.10 ± 0 |
| | Xiamenia | 0.05 ± 0.07 | 0.05 ± 0.07 | 0 ± 0 | 0 ± 0 |
| | Lactobacillus | 3.90 ± 4.95 | 0.60 ± 0.85 | 0.15 ± 0.07 | 6.50 ± 2.26* |
| | Ligilactobacillus | 0.25 ± 0.07 | 1.50 ± 1.56 | 7.75 ± 9.69 | 0.35 ± 0.07 |
| | Limosilactobacillus | 0.65 ± 0.78 | 0 ± 0 | 1.10 ± 1.41 | 3.60 ± 1.27*,# |
| | Streptococcus | 0 ± 0 | 0 ± 0 | 0.20 ± 0.28 | 0.10 ± 0.14 |
| | Flintibacter | 0.25 ± 0.21 | 0.50 ± 0.28 | 0.60 ± 0.42 | 0.80 ± 0.14* |
| | Intestinimonas | 0.40 ± 0.14 | 0.55 ± 0.49 | 0.25 ± 0.21 | 0.35 ± 0.07 |
| | Natranaerovirga | 0 ± 0 | 0 ± 0 | 0 ± 0 | 0.05 ± 0.07 |
| | Christensenella | 0.50 ± 0.57 | 1.80 ± 0* | 0.30 ± 0.28 | 0.20 ± 0.28 |
| | Guopingia | 0 ± 0 | 0.05 ± 0.07 | 0.05 ± 0.07 | 0.10 ± 0 |
| | Luoshenia | 0.10 ± 0.14 | 0.10 ± 0 | 0.10 ± 0 | 0.05 ± 0.07 |
| | Butyricicoccus | 0.20 ± 0.14 | 0.90 ± 0* | 0.15 ± 0.07 | 0.30 ± 0.28 |
| | Clostridium | 1.10 ± 0.99 | 0.65 ± 0.21 | 0.95 ± 0.78 | 0.65 ± 0.35 |
| | Hungatella | 0.85 ± 0.64 | 0.10 ± 0* | 0.15 ± 0.21 | 0.30 ± 0.28 |
| | Eubacterium | 0.40 ± 0 | 0.15 ± 0.21* | 0.20 ± 0.14# | 0.05 ± 0.07# |
| | Intestinibacillus | 0.15 ± 0.07 | 0.35 ± 0.35 | 0.25 ± 0.35 | 0.15 ± 0.07 |
| | Ihubacter | 0.05 ± 0.07 | 0 ± 0 | 0 ± 0 | 0 ± 0 |
| | Zhenpiania | 0 ± 0 | 0 ± 0 | 0.05 ± 0.07 | 0.05 ± 0.07 |
| | Feifania | 0.10 ± 0.14 | 0.05 ± 0.07 | 0.25 ± 0.07 | 0.10 ± 0* |
| | Gracilibacter | 0.05 ± 0.07 | 0.05 ± 0.07 | 0.05 ± 0.07 | 0.25 ± 0.35 |
| | Acetatifactor | 0.90 ± 0.57 | 2.55 ± 0.35* | 3.15 ± 3.04 | 1.40 ± 0.42 |
| | Anaerocolumna | 0.25 ± 0.21 | 0.25 ± 0.07 | 0.05 ± 0.07 | 0.70 ± 0.57* |
| | Anaeromicropila | 0.20 ± 0.28 | 0.15 ± 0.21 | 0 ± 0 | 0.10 ± 0 |
| | Anaerosporobacter | 0 ± 0 | 0.05 ± 0.07 | 0 ± 0 | 0.05 ± 0.07 |
| | Anaerotaenia | 0.15 ± 0.21 | 0.10 ± 0 | 0.20 ± 0.28 | 0.60 ± 0.14*,# |
| | Anaerotignum | 0.35 ± 0.21 | 0.20 ± 0.28 | 1.95 ± 2.05 | 0.65 ± 0.35 |
| | Blautia | 0.10 ± 0 | 0.10 ± 0.14 | 0.15 ± 0.07 | 0.25 ± 0.07* |

| Legend | Taxonomy | Distribution (%) | | | |
|---|---|---|---|---|---|
| | | Vehicle WT | MP WT | Vehicle Lep KO | MP Lep KO |
| | Coprococcus | 0.05 ± 0.07 | 0 ± 0 | 0.10 ± 0.14 | 0.15 ± 0.07 |
| | Dorea | 0 ± 0 | 0.35 ± 0.21* | 0.15 ± 0.07* | 0.20 ± 0.14* |
| | Eisenbergiella | 0.05 ± 0.07 | 1.10 ± 1.13 | 1.10 ± 0.71* | 0.55 ± 0.21* |
| | Enterocloster | 0.45 ± 0.21 | 0.05 ± 0.07* | 0.35 ± 0.49 | 0.90 ± 0.14# |
| | Faecalicatena | 0.05 ± 0.07 | 0.40 ± 0.42 | 0 ± 0 | 0.05 ± 0.07 |
| | Frisingicoccus | 0.30 ± 0.14 | 0 ± 0* | 0 ± 0* | 0.25 ± 0.35 |
| | Fusimonas | 0.20 ± 0 | 1.60 ± 0.71* | 1.60 ± 0.28* | 0.85 ± 0.78 |
| | Jingyaoa | 0.35 ± 0.35 | 0.40 ± 0.28 | 0.95 ± 0.78 | 0.90 ± 0.14* |
| | Jutongia | 1.20 ± 0.99 | 5.0 ± 7.07 | 0.30 ± 0 | 1.30 ± 0.71* |
| | Kineothrix | 3.35 ± 2.19 | 3.95 ± 0.64 | 8.80 ± 7.92 | 6.30 ± 0.99# |
| | Lachnoclostridium | 1.90 ± 0.85 | 4.40 ± 0.57* | 5.20 ± 0.42* | 4.32 ± 1.48# |
| | Lacrimispora | 2.65 ± 3.18 | 1.95 ± 1.06 | 1.05 ± 0.21 | 1.00 ± 1.27 |
| | Marvinbryantia | 0.05 ± 0.07 | 0 ± 0 | 0 ± 0 | 0.10 ± 0 |
| | Mediterraneibacter | 0.40 ± 0.14 | 1.15 ± 0.49* | 0.65 ± 0.07* | 0.65 ± 0.49 |
| | Muricomes | 0.10 ± 0.14 | 0.10 ± 0 | 0.05 ± 0.07 | 0.05 ± 0.07 |
| | Murimonas | 0.20 ± 0.28 | 0.15 ± 0.21 | 0.10 ± 0 | 0.60 ± 0.85 |
| | Qiania | 0 ± 0 | 0 ± 0 | 0.05 ± 0.07 | 0 ± 0 |
| | Roseburia | 0.20 ± 0.14 | 1.15 ± 1.20 | 2.30 ± 1.13* | 1.0 ± 0.71 |
| | Simiaoa | 0.05 ± 0.07 | 0.60 ± 0.42* | 1.75 ± 0.07* | 0.55 ± 0.64 |
| | Sporofaciens | 0.76 ± 0.49 | 2.45 ± 1.20* | 1.15 ± 0.35 | 1.0 ± 0.71 |
| | Tyzzerella | 0 ± 0 | 0.10 ± 0 | 0 ± 0 | 0 ± 0 |
| | Variimorphobacter | 0.05 ± 0.07 | 0.05 ± 0.07 | 0 ± 0 | 0 ± 0 |
| | Velocimicrobium | 2.05 ± 2.76 | 0 ± 0 | 0 ± 0 | 5.50 ± 7.78 |
| | Waltera | 0.05 ± 0.07 | 0.25 ± 0.07* | 0.05 ± 0.07 | 0.05 ±0.07 |
| | Wansuia | 0.05 ± 0.07 | 0.30 ± 0.42 | 0 ± 0 | 0.25 ± 0.21* |
| | Acetivibrio | 0 ± 0 | 0 ± 0 | 0 ± 0 | 0.05 ± 0.07 |
| | Acutalibacter | 0.05 ± 0.07 | 0.35 ± 0.07* | 0.10 ± 0.14 | 0.10 ± 0.14 |
| | Anaerobacterium | 0.05 ± 0.07 | 0 ± 0 | 0 ± 0 | 0.05 ± 0,07 |
| | Anaerotruncus | 0.20 ± 0 | 0.35 ± 0.07* | 0.05 ± 0.07* | 0.15 ± 0.07 |
| | Angelakisella | 0.15 ± 0.07 | 0.50 ± 0.28* | 0.45 ± 0.49 | 0.35 ± 0.07* |
| | Congzhengia | 0.10 ± 0.14 | 0 ± 0 | 0 ± 0 | 0.15 ± 0.21 |

| Legend | Taxonomy | Distribution (%) | | | |
|---|---|---|---|---|---|
| | | Vehicle WT | MP WT | Vehicle Lep KO | MP Lep KO |
| | Fumia | 0.05 ± 0.07 | 0 ± 0 | 0 ± 0 | 0.05 ± 0.07 |
| | Harryflintia | 0 ± 0 | 0.05 ± 0.07 | 0 ± 0 | 0 ± 0 |
| | Lawsonibacter | 0.75 ± 0.35 | 0.35 ± 0.21 | 0.40 ± 0.42 | 0.65 ± 0.78 |
| | Mageeibacillus | 0 ± 0 | 0.20 ± 0.28 | 0 ± 0 | 0.25 ± 0.07*,# |
| | Marasmitruncus | 0.05 ± 0.07 | 0.05 ± 0.07 | 0.15 ± 0.21 | 0.15 ± 0.07 |
| | Marseillibacter | 1.0 ± 0.28 | 1.05 ± 0.07 | 0.70 ± 0.71 | 1.20 ± 0.71 |
| | Neglectibacter | 0.05 ± 0.07 | 0.10 ± 0 | 0.05 ± 0.07 | 0.15 ± 0.07 |
| | Oscillibacter | 0.35 ± 0.21 | 1.05 ± 0.35* | 0.95 ± 0.49 | 1.05 ± 0.35# |
| | Paludihabitans | 0 ± 0 | 0 ± 0 | 0.05 ± 0.07 | 0 ± 0 |
| | Petroclostridium | 0 ± 0 | 0 ± 0 | 0 ± 0 | 0.05 ±0.07 |
| | Pseudoflavonifractor | 0.05 ± 0.07 | 0.05 ±0.07 | 0.10 ± 0.14 | 0.25 ± 0.07* |
| | Ruminiclostridium | 0.05 ± 0.07 | 0 ± 0 | 0.20 ± 0.28 | 0.15 ± 0.07 |
| | Ruminococcus | 0.95 ± 0.35 | 0.25 ± 0.35* | 0.15 ± 0.07* | 0.45 ± 0.07*,# |
| | Saccharofermentans | 0.25 ± 0.21 | 0.25 ± 0.21 | 0 ± 0* | 0.15 ± 0.07* |
| | Vescimonas | 0.80 ± 0.42 | 0.95 ± 0.07 | 0.85 ± 0.78 | 1.65 ± 0.92 |
| | Dehalobacterium | 0.15 ± 0.07 | 0.25 ± 0.07 | 0.15 ± 0.07 | 0.15 ± 0.07 |
| | Vallitalea | 1.20 ± 1.56 | 0.20 ± 0.28 | 0 ± 0 | 0.25 ± 0.35 |
| | Amedibacillus | 0 ± 0 | 0.05 ± 0.07 | 0 ± 0 | 0 ± 0 |
| | Faecalibaculum | 1.65 ± 2.19 | 1.80 ± 2.55 | 0 ± 0 | 0 ± 0 |
| | Longibaculum | 0.10 ± 0.14 | 0 ± 0 | 0 ± 0 | 0 ± 0 |
| | Tannockella | 0.05 ± 0.07 | 0 ± 0 | 0 ± 0 | 0 ± 0 |
| | Turicibacter | 2.25 ± 3.18 | 0 ± 0 | 6.35 ± 8.84 | 0 ± 0 |
| | Bacteroides | 4.05 ± 1.06 | 2.35 ± 1.20 | 4.05 ± 1.91 | 3.10 ± 0.57 |
| | Duncaniella | 10.25 ± 2.05 | 10.45 ± 4.03 | 6.05 ± 3.18 | 7.50 ± 2.26 |
| | Heminiphilus | 2.95 ± 2.76 | 0.35 ± 0.49 | 1.45 ± 0.64 | 0.20 ± 0.28* |
| | Muribaculum | 6.10 ± 0.28 | 5.75 ± 2.05 | 4.55 ± 0.64# | 4.60 ± 2.40 |
| | Paramuribaculum | 2.55 ± 0.78 | 2.40 ± 0.57 | 4.05 ± 0.49# | 3.20 ± 1.27 |
| | Sangeribacter | 2.25 ± 0.07 | 5.00 ± 0.85* | 1.90 ± 1.27 | 1.15 ± 0.21# |
| | Culturomica | 4.05 ± 0.49 | 7.85 ± 3.32 | 5.45 ± 5.16 | 4.95 ± 1.06 |
| | Prevotella | 4.75 ± 0.92 | 2.40 ± 2.26 | 1.50 ± 1.98# | 2.10 ± 2.40 |
| | Alistipes | 3.50 ± 0.42 | 9.40 ± 1.84* | 9.90 ± 2.12* | 11.35 ± 7.99 |

| Legend | Taxonomy | Distribution (%) | | | |
|---|---|---|---|---|---|
| | | Vehicle WT | MP WT | Vehicle Lep KO | MP Lep KO |
| | Tidjanibacter | 0.15 ± 0.07 | 0.05 ± 0.07 | 0.10 ± 0 | 0.15 ±0.07 |
| | Parabacteroides | 0.10 ± 0 | 0 ± 0 | 0.10 ± 0.14 | 0.05 ± 0.07 |
| | Helicobacter | 3.25 ± 4.17 | 2.90 ± 1.27 | 4.10 ±3.54 | 2.55 ± 3.61 |
| | Vampirovibrio | 0.15 ± 0.07 | 0.15 ± 0.07 | 0 ± 0* | 0.40 ± 0.14*,# |
| | Mucispirillum | 0.30 ± 0.28 | 0.15 ± 0.21 | 0 ± 0 | 1.10 ± 0.85 |
| | Anaeroplasma | 0.60 ± 0.85 | 1.95 ± 0.07* | 0.55 ± 0.35 | 3.10 ± 4.24 |
| | Aestuariispira | 0.30 ± 0.28 | 0.15 ± 0.21 | 0 ± 0 | 0.05 ± 0.07 |
| | Rhodospirillum | 0.25 ± 0.21 | 0.15 ± 0.21 | 0 ± 0* | 0.05 ± 0.07 |
| | Parasutterella | 0.20 ± 0.14 | 0.05 ± 0.07 | 0.10 ± 0.14 | 0.05 ± 0.07 |
| | Bilophila | 0.05 ± 0.07 | 0.10 ± 0 | 0.10 ± 0.14 | 0.20 ± 0* |
| | Desulfovibrio | 0.40 ± 0.28 | 0.15 ± 0.07 | 0.30 ± 0.42 | 0.40 ± 0.57 |
| | Akkermansia | 13.50 ± 9.48 | 0 ± 0* | 0 ± 0* | 0 ± 0* |
| | Others | 2.40 ± 1.27 | 3.0 ± 3.11 | 0.60 ± 0.42* | 1.05 ± 0.92 |

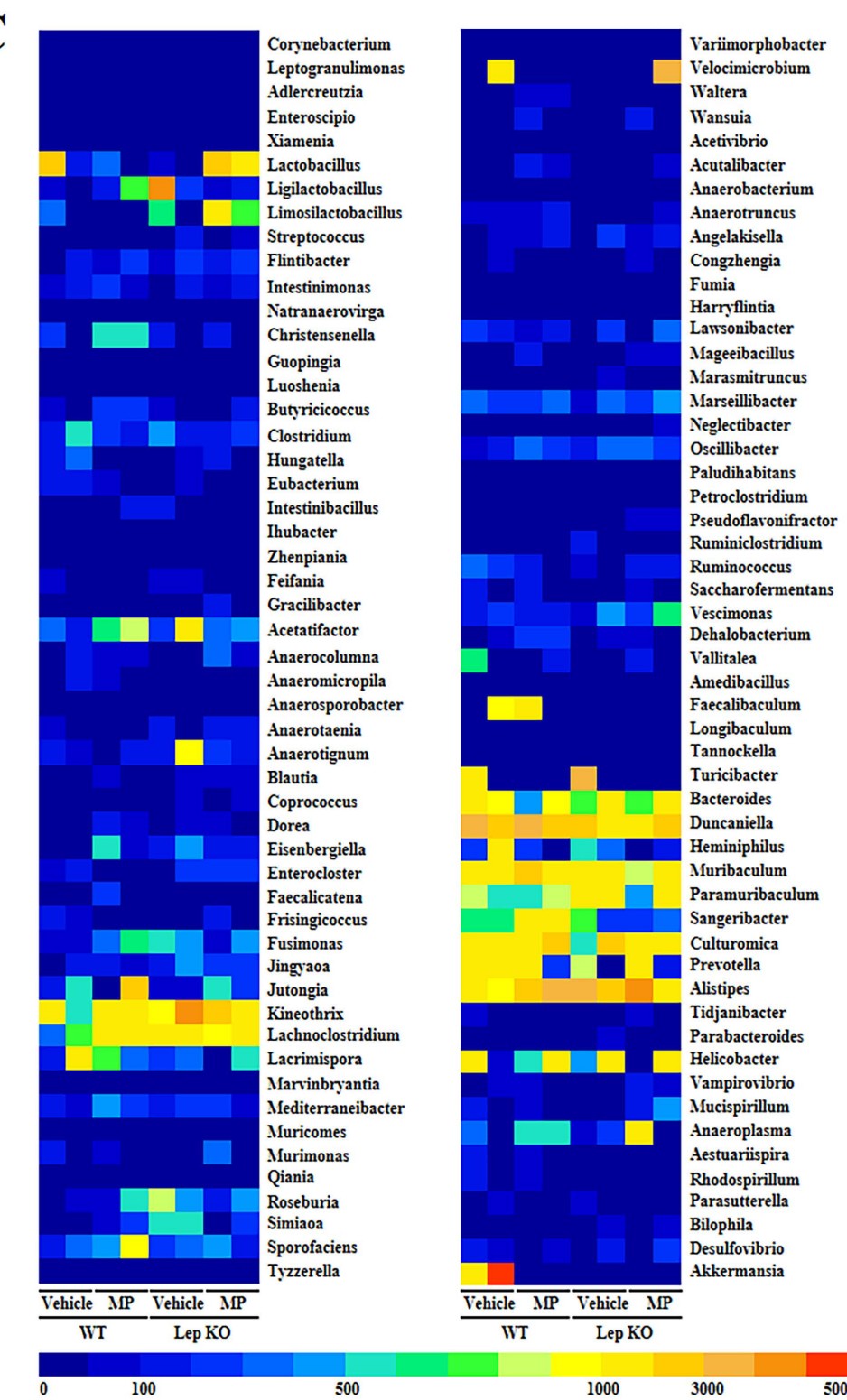

**Fig 3. Composition of the gut microbiota at the genus level following the administration of MP. (a)** Fecal microbiota distribution at the genus level in WT and Lep KO mice. **(b)** Relative level of fecal microbiota distribution at the genus level in WT and Lep KO mice. Each color represents one bacterial genus. **(c)** Heat map showing a significant difference. Different colors indicate the relative abundance of each genus. The data are reported as mean±SD values. *, $p < 0.05$ compared to the Vehicle-treated group. #, $p < 0.05$ compared to the WT mice. Abbreviation: WT; wild type, Lep KO; leptin knockout, MP; microplastics.

Lep KO mice (Fig 4A). The Shannon index increased in the same group after administering MP, but statistical significance was only detected in the MP-treated Lep KO mice (Fig 4B). The structural variations of the fecal microbiota were compared using PCoA with the Bray–Curtis dissimilarity matrix. Cluster separation was observed between the Vehicle-treated WT mice and MP-treated WT mice and between the Vehicle-treated Lep KO mice and MP-treated Lep KO mice in the unweighted and weighted UniFrac distance (Fig 5A and 5B). Therefore, the MP-caused hepatic metabolism disruption may be closely linked to changes in the structural diversity and dissimilarity of the fecal microbiota in Lep KO mice.

## Discussion

The changes in the composition of the gut microbiota, known as dysbiosis, have been implicated in various abnormal physical conditions, including chronic inflammatory bowel disease (IBD), diabetes, obesity, cardiovascular diseases, colorectal cancer, and depression [28]. On the other hand, no study has examined the effects of MP-caused hepatic metabolism disruption on the gut microbiota. This study analyzed the overall profile of the gut microbiota in Lep KO mice treated with MP for nine weeks. The present study revealed evidence that MP-caused hepatic metabolism abnormalities affect the gut microbiota differently in normal physiological conditions and obesity conditions of mice, even though further studies of these mechanisms of action are needed. However, it must be taken into account that changes in fecal microbiota of experimental animals may have the impact of various breeding environments including diet, drinking water, temperature, humidity, lighting cycle and microbial distribution.

Ob/ob mice with an impairment of leptin synthesis based on gene mutation were first reported as obese model at the Jackson Laboratory [29]. This model exhibits various obesity phenotypes, including excessive fat accumulation, unmanageable food intake, poor glucose tolerance, high insulin concentration, and transient hyperglycemia [30]. Recently, C57BL/6-Lep[em1hwl]/Korl mice with a deletion of the Lep gene were produced by the CRISPR-Cas9 system. They showed similar phenotypes, including obesity, hyperglycemia, hyperinsulinemia, liver damage markers, and hepatic steatosis, to the previous ob/ob model [31]. In particular, the oral administration of MP (0.4–0.6 µm) for nine weeks induced the disruption of lipid, glucose, and amino acid metabolism in the liver tissue of Lep KO mice [21]. In the MP-treated Lep KO mice of this study, the number of lipid droplets, NAFLD score and steatosis area were significantly reduced, contrary to expectations although the enhancements of them were detected in MP-treated WT mice [21]. However, the results of significant increase on these factors were detected in only ICR mice after MPs treatment with particle size of less than 50 µm, and HFD-induced obesity mouse model after treatment of an MPs [32–36]. These difference between Lep KO mice and HFD-induced obesity mice after MP treatment is thought to be related to the diversity of obesity-inducing mechanisms because some genetic and environmental factors are considered major causes of these diseases [21]. Especially, the previous study suggest that the fatty acid oxidation controlled by PPARα can be considered one of the key potential molecular mechanism involved in the impacts of MP on the hepatic lipid accumulation among four mechanism that consist of an uptake of circulating lipids, de novo lipogenesis (DNL), fatty acid oxidation (FAO), and an exportation of lipids [37,38]. Therefore, more studies on the effects of a MP-caused hepatic metabolism abnormality on the gut microbiota are required as a follow-up to previous studies because there are no reports of changes in the gut microbiota in Lep KO mice. But, this study provides the first scientific evidence for the disruption in gut microbiota in Lep KO mice treated with MP.

A disruption on MP-caused hepatic metabolism had been considered as a cause of the increasing populations of *Candidatus Melainabacteria* and Deferribacterota in Lep KO mice, as shown in Fig 2. *Candidatus Melainabacteria* is found widely in soil and water as well as in the human body, including the gut, respiratory tract, oral environments, and skin surface [39]. This phylum may play an important role in digesting a plat-rich diet or fiber in the human gut [40]. In addition, the members of family Deferribacteraceae are rod-shaped and Gram-negative bacteria and known for being strictly anaerobic and thermophilic. They are found in the intestinal mucus layer of animals and deep-sea hydrothermal vents [41,42]. Nevertheless, the relationship between the above two microbial phyla and MP exposure or obesity has not been studied until now.

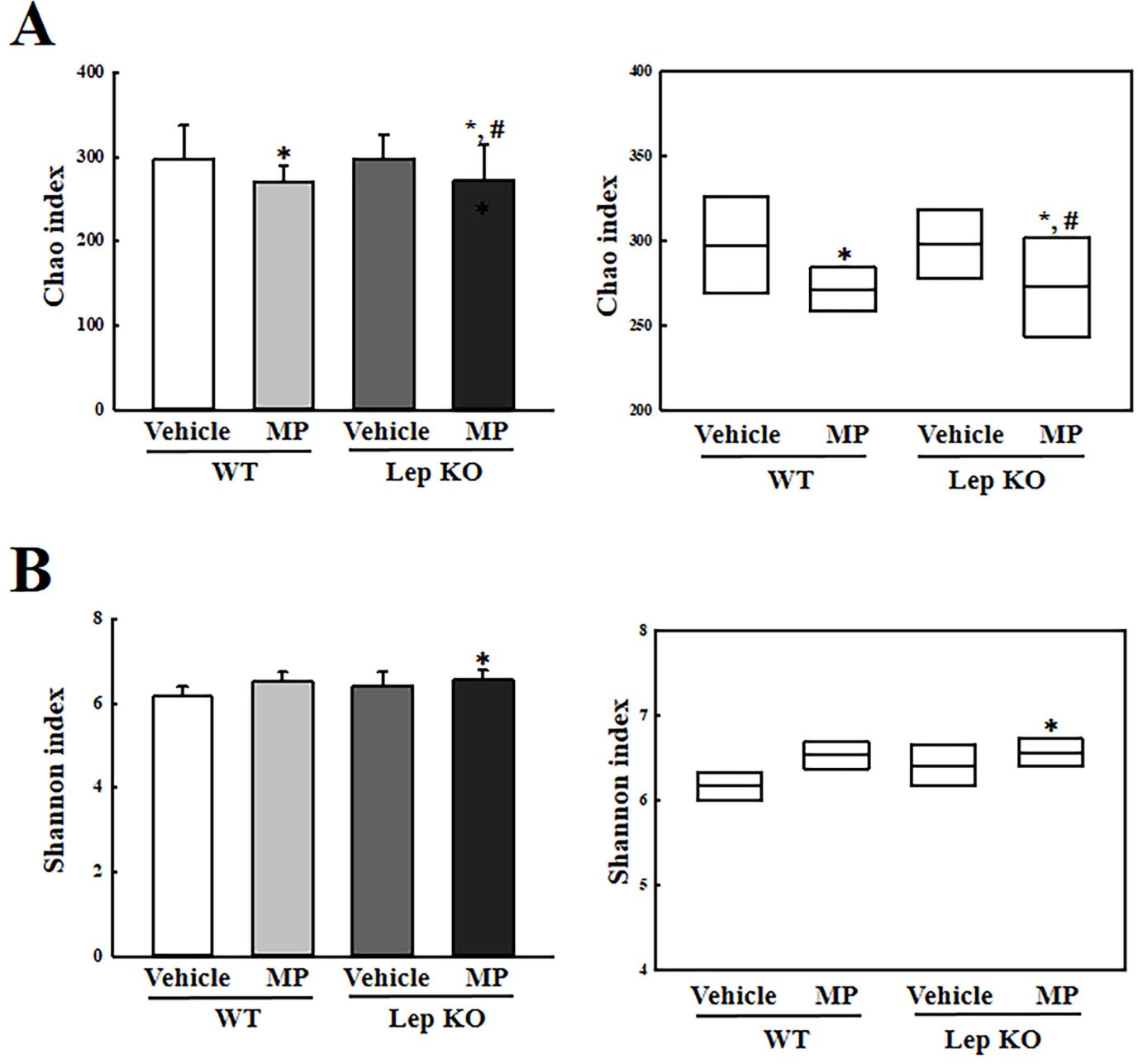

**Fig 4. Comparison of (a) Chao index and (b) Shannon diversity index of the fecal microbiota in WT and Lep KO mice.** The data are reported as mean±SD values. *, p<0.05 compared to the Vehicle-treated group. #, p<0.05 compared to the WT mice. Abbreviations: WT; wild type, Lep KO; leptin knockout, MP; microplastics.

Meanwhile, the present study provides the first evidence that changes in the *Candidatus Melainabacteria* and Defer-ribacterota populations may be linked to MP-caused hepatic metabolism abnormalities in Lep KO mice. The results of the present study were very different from previous studies that analyzed the changes in gut microbiota of normal mice after administering MP. This difference was attributed to the obesity condition or the size of the PS-MP administered. Furthermore, in all previous studies, the changes in the gut microbiota at the phylum level were very diverse, except for

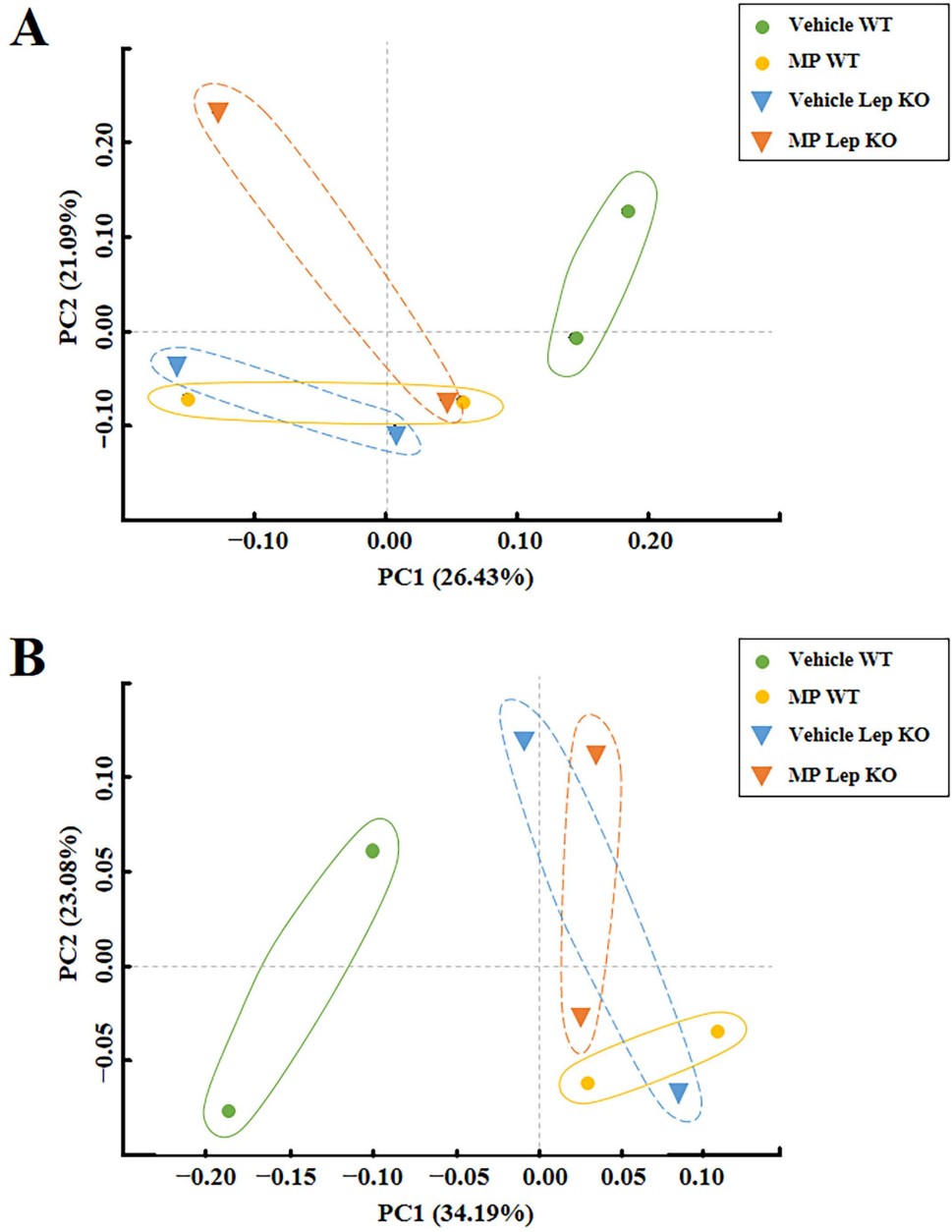

**Fig 5. PCoA plot analysis for dissimilarity. (a)** Unweighted unifrac. **(b)** Weighted unifrac. The PCoA focused on fecal bacterial communities using the principle components in WT and Lep KO mice. The spatial distance measure indicates the degree of similarity of bacterial taxa in the fecal sample. Abbreviations: PCoA; Principal Coordinates Analysis, WT; wild type, Lep KO; leptin knockout, MP; microplastics.

a few similar phyla. The Firmicutes/Bacteroidetes ratio increased or decreased in C57BL/6 mice after treatment with 20 nm–5 µm PS-MP [12,16,18]. Proteobacteria and Actinobacteriota were identified as commonly changing phyla after administering PS-MP, despite there are few differences in administration conditions and changes in the population ratios [10,11,17,18]. Moreover, Bacteroidota was significantly changed in the C57BL/6 mice treated with PS-MP, while similar changes with Firmicutes were detected in ICR or C57BL/6 mice treated with PS-MP [11,13,14,19].

ICR, C57BL/6, and BALB/c mice showed various changes in the gut microbiota at the genus level after administering PS-MP of various sizes. Nevertheless, few genera showed common changes. Among them, *Parabacteroides* showed significant changes in ICR mice treated with 5 μm PS-MP for five or six weeks as well as in C57BL/6 mice administrated orally with 0.5 and 5 μm PS-MP for eight weeks and 0.1, 1, and 5 μm for four weeks [10,11,16,20]. Although this genus has been associated with beneficial effects on human health, it decreased in most cases and only increased in one case after a PS-MP treatment [43]. In addition, *Bacteroides* was identified as a changing genus after the PS-MP treatment. They decreased in C57BL/6 mice administered 80 nm, 5 μm, and 10 μm orally for 42 days or 0.5 and 5 μm for eight weeks, while the mice cotreated with PS-MP (0.1, 5, and 50 μm) and dextran sulfate sodium (DSS) for four weeks showed an increasing pattern [16,19,20]. *Bacteroides* play a beneficial role, including nutrient sources, carbohydrate metabolism, sugar transport, and environment sensing, and a harmful role, such as virulence, bacterial capsule, enterotoxin, and endotoxin in human health [44]. *Prevotella* and *Dehalobacterium* showed opposite changes in ICR mice after the PS-MP treatment. They were increased by treatment with 0.5 and 5 μm for five weeks but decreased by 5 μm for six weeks [10,11]. Furthermore, significant changes in the *Ruminococcus* population were detected in the 0.5 and 5 μm PS-MP-treated ICR and C57BL/6 mice, as well as in the 80 nm, 5 μm, and 10 μm PS-MP-treated C57BL/6 mice [10,16,19]. The abundance of this genus decreased in various diseases, including inflammatory bowel disease, Parkinson's disease, and amyotrophic lateral sclerosis. *Ruminococcus gnavus* is associated with Crohn's disease [45–47]. The present study characterized the gut microbiota profile at the genus level in PS-MP-administered Lep KO mice with hepatic metabolism abnormalities. The administration of PS-MP for nine weeks induced significant changes in 12 genera. Among them, *Lactobacillus* and *Ruminococcus* were increased significantly in the PS-MP-treated Lep KO mice. Changes in these genera have been also observed in previous studies. *Lactobacillus* was remarkably changed in the PS-MP (<5 mm)-treated C57BL/6 mice for 30 days, MP (0.1, 5, and 10 μm) and DSS-cotreated C57BL/6 mice for four weeks, and PS-MP (100 nm and 1 μm)-treated C57BL/6 mice for 60 days [15,17,20]. *Ruminococcus* increased after treatment with 0.5 and 5 μm PS-MP and decreased after treatment with 80 nm, 5 μm, and 10 μm PS-MP [10,16,19]. The results of the present study provide the first evidence for the microbiota profile at the genus level in MP-treated Lep KO mice with hepatic metabolism abnormality. Among microorganisms changed by MP treatment, *Firmicutes* and *Bacteroidetes* was closely associated with obesity. Obese patients showed the imbalance of microbiota with increased *Firmicutes* and decreased *Bacteroidetes*, and these changes exacerbate an obesity and metabolic diseases through stimulating inflammation, energy metabolism changes, and fat accumulation [48,49].

Generally, the scientific verification of causality for biological process requires an analysis of timativity, specificity, dose-reactivityitamins, and reversibility [50]. But, this study did not fully demonstrate causality between MP-induced hepatic disruption and gut microbiota changes although it shows the correlation between them. The results of our study were provided the evidences for the temporality and specificity between them. The temporality were demonstrated by the sequential change in the microbiota following disturbance of the liver, while specificity was proven to be a major changes in only manipulations targeting the liver pathway as described in previous studies [51,52]. However, our results do not present evidence for dose-reactivity and reversibility between them. To demonstrate the dose-reactivity, it is necessary to further analyze the dose-dependent response between MP-induced liver damages and changes of microbiota. Also, the reversibility can be demonstrated by whether recovery of liver damage is linked to recovery of microbiota composition.

## Conclusions

This study characterized the gut microbiota profile in Lep KO mice with hepatic metabolism abnormalities after the oral administration of PS-MP for nine weeks. These results show that the populations of *Candidatus Melainabacteria* and Deferribacterota were identified as gut microbiota with significant changes at the phylum level, while 12 genera, including *Lactobacillus*, were detected at the genus level in PS-MP-treated Lep KO mice. In addition, these results suggest that the hepatic metabolism disruption caused by MP administration for nine weeks may be closely linked to changes in the

structural diversity and dissimilarity of the fecal microbiota in Lep KO mice. Nevertheless, this study had a few limitations on the verification experiment. The present study did not examine whether the identified microorganisms are directly related to the abnormal liver metabolism of Lep KO mice. Therefore, further studies will be needed to determine the role of a single microorganism identified from this study in the antibiotics-induced depletion of microbiota (AiDM) model through fecal microbiota transplantation. Also, six mice per group used for microbiota analyses in our study can be considered as another limitations to be small in sample size although fecal samples were pooled to prevent statistical evaluation of individual variants and to limit the robustness of diversity and abundance comparisons. Furthermore, the metagenomic or metabolomic analyses are needed as a further study to link microbiota changes with functional metabolic consequences.

## Supporting information

**S1 File. Supporting information.**
(ZIP)

## Acknowledgments

We thank Jin Hyang Hwang, the animal technician, for directing the animal care and use at the Laboratory Animal Resources Center at Pusan National University.

## Author contributions

**Conceptualization:** Yu Sang Choi, Yu Jeong Roh, Dae Youn Hwang.

**Data curation:** Yu Sang Choi, Yu Jeong Roh, Ye Eun Ryu, Ye Ryeong Kim, Hyun Su Park.

**Formal analysis:** Yu Sang Choi, Yu Jeong Roh, Ji Eun Kim, Hee Jin Song, Ayun Seol.

**Funding acquisition:** Dae Youn Hwang.

**Investigation:** Yu Sang Choi, Yu Jeong Roh.

**Methodology:** Yu Sang Choi, Yu Jeong Roh.

**Project administration:** Dae Youn Hwang.

**Resources:** Hong Joo Son, Yu Jin Kim.

**Software:** Ji Eun Kim, Hee Jin Song, Ayun Seol.

**Supervision:** Dae Youn Hwang.

**Validation:** Su Jeong Lim, Su Ha Wang, Ji Eun Sung.

**Visualization:** Yu Sang Choi, Yu Jeong Roh.

**Writing – original draft:** Dae Youn Hwang.

**Writing – review & editing:** Yu Sang Choi, Yu Jeong Roh, Eun Sang Jung, Sunho Park.

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
