## [Decision Letter · Decision Letter 0]

20 Aug 2025

Dear Dr. Hwang,

Thank you for submitting your manuscript to PLOS ONE. After careful consideration, we feel that it has merit but does not fully meet PLOS ONE’s publication criteria as it currently stands. Therefore, we invite you to submit a revised version of the manuscript that addresses the points raised during the review process.

We look forward to receiving your revised manuscript.

Kind regards,

Sayed Haidar Abbas Raza

Academic Editor

PLOS ONE

Journal Requirements:

4. We note that your Data Availability Statement is currently as follows: All relevant data are within the manuscript and in Supporting Information files.

6. Please ensure that you refer to Figure 5 in your text as, if accepted, production will need this reference to link the reader to the figure.

Reviewers' comments:

Reviewer's Responses to Questions

**Comments to the Author**

1. Is the manuscript technically sound, and do the data support the conclusions?

Reviewer #1: Yes

Reviewer #2: Partly

2. Has the statistical analysis been performed appropriately and rigorously?

Reviewer #1: Yes

Reviewer #2: No

3. Have the authors made all data underlying the findings in their manuscript fully available?

Reviewer #1: Yes

Reviewer #2: Yes

4. Is the manuscript presented in an intelligible fashion and written in standard English?

Reviewer #1: Yes

Reviewer #2: Yes

Reviewer #1: The study is well-designed, presents novel findings, and is clearly written. Methodology is sound, and results are supported by data. I recommend publication without major revisions. I am happy to endorse the manuscript.

Reviewer #2: This study investigates whether long-term oral exposure to polystyrene microplastics (PS-MP) alters gut microbiota composition in leptin-deficient (Lep KO) mice with known hepatic metabolism disruption. WT and Lep KO mice were treated with PS-MP or vehicle for nine weeks, after which liver histology and 16S rRNA sequencing of fecal samples were performed. MP-treated Lep KO mice showed MP accumulation in liver tissue and reduced hepatic steatosis. Microbiota profiling revealed significant shifts at both the phylum and genus levels, with Lep KO mice showing marked increases in Candidatus Melainabacteria, Deferribacterota, and several genera including Lactobacillus and Ruminococcus. Alpha diversity (Chao1) decreased in both MP-treated WT and Lep KO mice, while Shannon diversity increased only in MP-treated Lep KO mice. PCoA analysis showed clear clustering differences between treatment groups. The authors conclude that MP-induced hepatic metabolism disruption is associated with dysbiosis of gut microbiota in obese Lep KO mice. The manuscript is well written but I have a few concerns

1. Sample size is small with only six mice per group for microbiota analysis. Fecal samples were pooled, preventing statistical assessment of individual variation and limiting the robustness of diversity and abundance comparisons.

2. The study shows correlations between MP-induced hepatic disruption and gut microbiota changes but does not establish causality.

3. Reduction in lipid droplets/steatosis in MP-treated Lep KO mice contradicts expectations and is not sufficiently explained mechanistically.

4. 16S rRNA sequencing provides taxonomic shifts, but no metagenomic or metabolomic analysis to link microbiota changes with functional metabolic consequences.

5. The relevance of specific taxa changes (e.g., large Lactobacillus increase) to liver metabolism or obesity is not well integrated into the discussion.

6. Housing conditions, cage effects, and diet could influence microbiota, but these are not discussed or statistically adjusted for.

7. Some key data (e.g., individual animal data points, variability measures for pooled samples) are missing.

8. MP accumulation in liver shown qualitatively by fluorescence microscopy but not quantified to correlate with microbiota changes.

**Do you want your identity to be public for this peer review?** For information about this choice, including consent withdrawal, please see our Privacy Policy

Reviewer #1: No

Reviewer #2: No

---

## [Author Response · Author response to Decision Letter 1]

7 Oct 2025

Journal Requirements :

☞ According to your comments, the manuscript has been corrected to match the PLOS ONE style.

☞ They have been deleted from manuscript.

☞ We have corrected to include the gran number.

4. We note that your Data Availability Statement is currently as follows: All relevant data are within the manuscript and in Supporting Information files.

☞ According to your comments, the manuscript has been corrected to match the PLOS ONE style.

☞ Above issue have been already described in Material and methods section (line 81-83) as follows;

“The Pusan National University-Institutional Animal Care and Use Committee (PNU-IACUC) approved the protocols of the animal experiments based on ethical procedures for scientific care (Approval Number PNU-2022-0191 and PNU-2023-0350).”

6. Please ensure that you refer to Figure 5 in your text as, if accepted, production will need this reference to link the reader to the figure.

☞ This is our mistake. We have citied Figure 5 in Results section (line 218-221) as follows;

“Cluster separation was observed between the Vehicle-treated WT mice and MP-treated WT mice and between the Vehicle-treated Lep KO mice and MP-treated Lep KO mice in the unweighted and weighted UniFrac distance (Figure 5A and B).”

☞ Thank you for your advice.

Reviewer #1 :

The study is well-designed, presents novel findings, and is clearly written. Methodology is sound, and results are supported by data. I recommend publication without major revisions. I am happy to endorse the manuscript.

☞ Thank you very much.

Reviewer #2 :

This study investigates whether long-term oral exposure to polystyrene microplastics (PS-MP) alters gut microbiota composition in leptin-deficient (Lep KO) mice with known hepatic metabolism disruption. WT and Lep KO mice were treated with PS-MP or vehicle for nine weeks, after which liver histology and 16S rRNA sequencing of fecal samples were performed. MP-treated Lep KO mice showed MP accumulation in liver tissue and reduced hepatic steatosis. Microbiota profiling revealed significant shifts at both the phylum and genus levels, with Lep KO mice showing marked increases in Candidatus Melainabacteria, Deferribacterota, and several genera including Lactobacillus and Ruminococcus. Alpha diversity (Chao1) decreased in both MP-treated WT and Lep KO mice, while Shannon diversity increased only in MP-treated Lep KO mice. PCoA analysis showed clear clustering differences between treatment groups. The authors conclude that MP-induced hepatic metabolism disruption is associated with dysbiosis of gut microbiota in obese Lep KO mice. The manuscript is well written but I have a few concerns

1. Sample size is small with only six mice per group for microbiota analysis. Fecal samples were pooled, preventing statistical assessment of individual variation and limiting the robustness of diversity and abundance comparisons.

☞ This is good point. Actually, the number of animal per group has been considered as key factor to ensure the reliability of the data from the animal study. However, since analyzing samples obtained from a large number of animals requires a large economic cost, most papers commonly use the method of pooling fecal samples. Please understand this situation.

Meanwhile, to address above issues, we have added the justification for the total number of animals in Materials and methods section (line 102-105) as follows;

“To ensure the reliability of the data from the animal study, the total number of animals was determined as 24 using G‑POWER 3.1.9.7 (Heinrich‑Heine‑Universität Düsseldorf, Germany) with the α error probability of 0.05, effect size of 0.9 and a power of 0.95.”

In addition, the problem of data analysis due to the pooling of samples was added to the Conclusion (line 336-340) as a limitation of the study as follows;

“Also, six mice per group used for microbiota analyses in our study can be considered as another limitations to be small in sample size although fecal samples were pooled to prevent statistical evaluation of individual variants and to limit the robustness of diversity and abundance comparisons.”

2. The study shows correlations between MP-induced hepatic disruption and gut microbiota changes but does not establish causality.

☞ According to your comments, we have further described the paragraph for causality in Discussion (line 313-323) as follows;

“Generally, the scientific verification of causality for biological process requires an analysis of timativity, specificity, dose-reactivityitamins, and reversibility [50]. But, this study did not fully demonstrate causality between MP-induced hepatic disruption and gut microbiota changes although it shows the correlation between them. The results of our study were provided the evidences for the temporality and specificity between them. The temporality were demonstrated by the sequential change in the microbiota following disturbance of the liver, while specificity was proven to be a major changes in only manipulations targeting the liver pathway as described in previous studies [51,52]. However, our results do not present evidence for dose-reactivity and reversibility between them. To demonstrate the dose-reactivity, it is necessary to further analyze the dose-dependent response between MP-induced liver damages and changes of microbiota. Also, the reversibility can be demonstrated by whether recovery of liver damage is linked to recovery of microbiota composition.”

3. Reduction in lipid droplets/steatosis in MP-treated Lep KO mice contradicts expectations and is not sufficiently explained mechanistically.

☞ According to your comments, we have further described the paragraph for contradicts expectations in Discussion (line 244-254) as follows;

“In the MP-treated Lep KO mice of this study, the number of lipid droplets, NAFLD score and steatosis area were significantly reduced, contrary to expectations although the enhancements of them were detected in MP-treated WT mice [21]. However, the results of significant increase on these factors were detected in only ICR mice after MPs treatment with particle size of less than 50 �m, and HFD-induced obesity mouse model after treatment of an MPs [32-36]. These difference between Lep KO mice and HFD-induced obesity mice after MP treatment is thought to be related to the diversity of obesity-inducing mechanisms because some genetic and environmental factors are considered major causes of these diseases [21]. Especially, the previous study suggest that the fatty acid oxidation controlled by PPARα can be considered one of the key potential molecular mechanism involved in the impacts of MP on the hepatic lipid accumulation among four mechanism that consist of an uptake of circulating lipids, de novo lipogenesis (DNL), fatty acid oxidation (FAO), and an exportation of lipids [37,38].”

4. 16S rRNA sequencing provides taxonomic shifts, but no metagenomic or metabolomic analysis to link microbiota changes with functional metabolic consequences.

☞ This is good point. As you suggestion, it is important and ideal to link microbiota changes to functional metabolic consequences. However, we could not attempt these analyses in this study because they require a lot of time and economic costs. Please understand this situation.

Therefore, these issues have been further described as limitation of our study in Conclusion (line 340-341) as follows;

“Furthermore, the metagenomic or metabolomic analyses are needed as a further study to link microbiota changes with functional metabolic consequences.”

5. The relevance of specific taxa changes (e.g., large Lactobacillus increase) to liver metabolism or obesity is not well integrated into the discussion.

☞ According to your comments, we have further described the paragraph for above issues in Discussion (line 308-312) as follows;

“Among microorganisms changed by MP treatment, Firmicutes and Bacteroidetes was closely associated with obesity. Obese patients showed the imbalance of microbiota with increased Firmicutes and decreased Bacteroidetes, and these changes exacerbate an obesity and metabolic diseases through stimulating inflammation, energy metabolism changes, and fat accumulation [48,49].”

6. Housing conditions, cage effects, and diet could influence microbiota, but these are not discussed or statistically adjusted for.

☞ This is good point. According to your comments, we have further described the paragraph for above issues in Discussion (line 232-235) as follows;

“However, it must be taken into account that changes in fecal microbiota of experimental animals may have the impact of various breeding environments including diet, drinking water, temperature, humidity, lighting cycle and microbial distribution.”

7. Some key data (e.g., individual animal data points, variability measures for pooled samples) are missing.

☞ According to your comments, we have uploaded the some key data as Supporting Information files based on the request of the Editorial Office.

8. MP accumulation in liver shown qualitatively by fluorescence microscopy but not quantified to correlate with microbiota changes.

☞ According to your comments, we have added the data of MP accumulation into Figure 1B as following;

---

## [Decision Letter · Decision Letter 1]

28 Oct 2025

Dysbiosis of gut microbiota in C57BL/6-Lepem1hwl/Korl mice during microplastics-caused hepatic metabolism disruption

PONE-D-25-33830R1

Dear Dr. Hwang,

We’re pleased to inform you that your manuscript has been judged scientifically suitable for publication and will be formally accepted for publication once it meets all outstanding technical requirements.

Kind regards,

Sayed Haidar Abbas Raza, PhD

Academic Editor

PLOS ONE

Additional Editor Comments (optional):

Reviewers' comments:

Reviewer's Responses to Questions

**Comments to the Author**

Reviewer #1: All comments have been addressed

Reviewer #2: All comments have been addressed

2. Is the manuscript technically sound, and do the data support the conclusions?

Reviewer #1: Yes

Reviewer #2: Yes

3. Has the statistical analysis been performed appropriately and rigorously?

Reviewer #1: Yes

Reviewer #2: Yes

4. Have the authors made all data underlying the findings in their manuscript fully available?

Reviewer #1: Yes

Reviewer #2: Yes

5. Is the manuscript presented in an intelligible fashion and written in standard English?

Reviewer #1: Yes

Reviewer #2: Yes

Reviewer #1: The authors have sufficiently addressed all the aspects of the previous concerns. They have updated the manuscript with the necessary changes. I am happy to endorse the paper.

Reviewer #2: This revised manuscript investigates how long-term oral exposure to polystyrene microplastics (PS-MP) influences gut microbiota composition in leptin-deficient (Lep KO) mice with disrupted hepatic metabolism. The study integrates histopathological liver analyses and 16S rRNA-based microbiota profiling to explore the gut–liver axis under microplastic-induced metabolic stress.

The authors have addressed nearly all concerns raised in the previous review round. The revisions substantially improve the overall rigor and completeness of the work.

**Do you want your identity to be public for this peer review?** For information about this choice, including consent withdrawal, please see our Privacy Policy

Reviewer #1: No

Reviewer #2: No

---

## [Editor Report · Acceptance letter]

PONE-D-25-33830R1

PLOS ONE

Dear Dr. Hwang,

I'm pleased to inform you that your manuscript has been deemed suitable for publication in PLOS ONE. Congratulations! Your manuscript is now being handed over to our production team.

Kind regards,

on behalf of

Dr. Sayed Haidar Abbas Raza

Academic Editor

PLOS ONE